# High-Resolution Indicators of Soil Microbial Responses to N Fertilization and Cover Cropping in Corn Monoculture

Nakian Kim [1], Chance W. Riggins [1], María C. Zabaloy [2], Marco Allegrini [3], Sandra L. Rodriguez-Zas [4] and María B. Villamil [1,*]

[1] Department of Crop Sciences, University of Illinois, Urbana, IL 61801, USA; nakhyun2@illinois.edu (N.K.); cwriggin@illinois.edu (C.W.R.)
[2] Centro de Recursos Naturales Renovables de la Zona Semiárida (CERZOS, UNS-CONICET), Departamento de Agronomía, Universidad Nacional del Sur, Bahía Blanca B8000, Argentina; mzabaloy@uns.edu.ar
[3] Laboratorio de Biodiversidad Vegetal y Microbiana, Instituto de Investigaciones en Ciencias Agrarias Rosario (IICAR-CONICET), Universidad Nacional de Rosario, Zavalla S2000, Argentina; marcalleg88@hotmail.com
[4] Department of Animal Sciences, University of Illinois, Urbana, IL 61801, USA; rodrgzzs@illinois.edu
* Correspondence: villamil@illinois.edu

**Abstract:** Cover cropping (CC) is the most promising in-field practice to improve soil health and mitigate N losses from fertilizer use. Although the soil microbiota play essential roles in soil health, their response to CC has not been well characterized by bioindicators of high taxonomic resolution within typical agricultural systems. Our objective was to fill this knowledge gap with genus-level indicators for corn [*Zea mays* L.] monocultures with three N fertilizer rates (N0, N202, N269; kg N ha⁻¹), after introducing a CC mixture of cereal rye [*Secale cereale* L.] and hairy vetch [*Vicia villosa* Roth.], using winter fallows (BF) as controls. A 3 × 2 split-plot arrangement of N rates and CC treatments was studied in a randomized complete block design with three replicates over two years. Bacterial and archaeal 16S rRNA and fungal ITS regions were sequenced with Illumina MiSeq system. Overall, our high-resolution bioindicators were able to represent specific functional or ecological shifts within the microbial community. The abundances of indicators representing acidophiles, nitrifiers, and denitrifiers increased with N fertilization, while those of heterotrophic nitrifiers, nitrite oxidizers, and complete denitrifiers increased with N0. Introducing CC decreased soil nitrate levels by up to 50% across N rates, and CC biomass increased by 73% with N fertilization. CC promoted indicators of diverse functions and niches, including N-fixers, nitrite reducers, and mycorrhizae, while only two N-cycling genera were associated with BF. Thus, CC can enhance the soil biodiversity of simplified cropping systems and reduce nitrate leaching, but might increase the risk of nitrous oxide emission without proper nutrient management. This primary information is the first of its kind in this system and provided valuable insights into the limits and potential of CC as a strategy to improve soil health.

**Keywords:** bioinformatics; nitrogen cycling; soil microbiota; maize; nitrate leaching; nitrous oxide emission

## 1. Introduction

Soil health represents a soil's capacity for ecosystem services, making it a crucial component of sustainable agriculture [1,2]. Hence, the soil health of critical agricultural regions such as the US Midwest must be protected to maintain global food security [3–6]. However, this region is dominated by simplified and intensely managed cropping systems centered on corn [*Zea mays* L.] and soybean [*Glycine max* (L.) Merr.], which makes the soil more vulnerable to both anthropogenic and natural disturbances [7–9]. Corn-based systems often have low N use efficiency, leading to excess soil N [10] that causes a

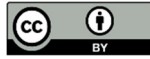

chemical imbalance that degrades soil health and contributes to greenhouse gas (GHG) emissions and nutrient pollutions [11–14].

Cover cropping (CC) has many potential benefits for soil health, such as providing physical protections, adding organic matter, and scavenging excess soil N [3,15,16]. Thus, CC has been proposed as a tool of ecological intensification to improve soil health [17,18]. In particular, CC is anticipated for its ability to mitigate soil chemical imbalances by immobilizing excess soil nutrients as biomass and releasing them slowly through decomposition. Indeed, past primary research and research syntheses demonstrated that CC that includes non-legumes significantly reduces $NO_3^-$ leaching [3,16]. Yet, many aspects of CC interaction with the soil remain unexplored or ambiguous. For example, whether CC can effectively reduce emissions of nitrous oxide ($N_2O$), a very potent GHG, may depend on soil microbial responses to CC management [19]. Thus, the soil microbes can regulate the CC impacts on soil health as the fundamental driver of soil processes [15,20,21]. Likewise, a diverse microbial community with groups of overlapping roles leads to higher functional redundancy that indicates a healthy and resilient soil [22]. Therefore, proper evaluation of CC as a tool to alleviate soil health degradation and N loss requires a better understanding of the soil biodiversity under CC management.

Despite the importance of soil biodiversity for successful CC strategies, many gaps in knowledge still exist due to the vast complexity of the soil microbiome. Therefore, indicators need to be identified to describe a microbiome reliably. Initially, properties of the whole microbial community, such as total microbial biomass, respiration, and $\alpha$-diversity, served as indicators in CC research [15,23–26]. For example, a global meta-analysis by Kim et al. [25] found beneficial effects of CC on thirteen parameters of microbial abundance, activity, and diversity. However, these indicators from broad-scale integrative methods do not describe the microbial diversity and functionality with enough detail. Therefore, CC research adopted approaches, such as metabarcoding, that can quantify the changes in the abundances of individual microbial taxa and identify those sensitive to the treatments as a type of bioindicator [27]. These indicator taxa can provide taxonomic characterization of the responsive microbes that can complement other bioindicators such as β-diversity and the functional genes. This effort started from identifying the taxa sensitive to CC at lower taxonomic resolutions [28]. For example, a study by Castle et al. [29] showed that the N fertilization rate changed the relative abundances of bacterial phyla, while CC proved to be a stronger predictor of fungal community composition [29]. However, due to the wide ecological and functional diversity within each phylum, these bioindicators are still too low in taxonomic resolution to infer on more specific microbial processes, such as plant symbiosis or a particular step of denitrification [30].

The recent advancements in sequencing technology and bioinformatics have enabled the identification of genus- or even species-level indicators from surveying the vast microbial community data at higher taxonomic resolutions. Thus, these indicator taxa can represent more specific microbial guilds or functions. For example, Villamil et al. [31] used bacterial and archaeal 16S rRNA and fungal internal transcribed spacer (ITS) sequence data to select genus-level indicators through predictor screening and principal component analysis. They found indicator genera whose responses to management practices were consistent with their known characteristics, such as those of acidophiles that increased with soil acidification. Thus, the authors demonstrated that low-rank indicator taxa can reliably describe the soil microbial community and complement other taxonomic or functional bioindicators. Therefore, identifying genus-level bioindicators after introducing CC may lead to detailed insights into whether CC can improve the soil biodiversity. So far, however, only a few studies have identified the high taxonomic resolution indicators of CC. Alahmad et al. [32] identified species-level indicators to represent guilds that specialize in different C substrate groups, but their system widely differed from the simple cropping systems. Another study by Kim et al. [33] identified indicator genera within a typical corn–soybean rotation after five years of CC. However, this study did not include an unfertilized control to test the N rate effect and included tillage treatments in the model.

Thus, high-resolution indicator taxa have not been well-identified for CC deployment in simplified cropping systems with and without N fertilization.

Therefore, our objective was to describe the soil microbial community of a typical corn monoculture with and without N fertilizers upon introducing CC with high taxonomic resolution, using bacterial, archaeal, and fungal genera as indicators. Our aim was to use these bioindicators to investigate whether CC can increase soil biodiversity and has the potential to improve soil health of this system. Three published studies have each characterized the soil properties [14], the N-cycling genes [21], and the indicator genera [31] of the experimental site of this study before CC deployment. Consistent with these past studies, we expected to still observe soil acidification and an increased abundance of acidophiles, nitrifiers, and denitrifiers with N fertilization. Thus, based on the assumption that CC will improve the soil biodiversity and N use efficiency of corn monocultures, we hypothesized that CC would (1) reduce soil $NO_3^-$ levels through assimilation, which would (2) compete with the denitrifiers for $NO_3^-$ and decrease their abundances, and that (3) indicators associated with CC would represent more diverse niches and functions than those of bare fallow. These bioindicators will help us assess whether CC can improve the soil health of simplified and intensely managed cropping systems and reduce their soil N loss. In addition, these bioindicators will facilitate identifying the core microbiota that could be managed to optimize CC in high-N-input agroecosystems.

## 2. Materials and Methods

### 2.1. Experimental Site Description and Management Practices

The field experimental site was established in 1981 at the Northwestern Illinois Agricultural Research and Demonstration Center (40°55′50″ N, 90°43′38″ W) to study the effects of N fertilization rates on corn yields when the crop is in a corn monoculture or short rotation with soybeans (Figure S1). The site has mean annual precipitation and temperature of 914 mm and 10.6 °C, respectively [34]. The soil series is Muscatune silt loam (fine-silty, mixed, mesic Aquic Argiudoll) on nearly flat topography [35]. These are dark-colored and very deep soils with moderate permeability and low surface runoff potential developed under prairie vegetation in a layer of loess 2–3 m thick over glacial till [35]. Further information regarding the experimental site and management before 2018 can be found in Kim et al. [14].

This study centers on the introduction of CC into the continuous corn management plots that spanned two CC growing seasons: 2018–2019 and 2019–2020. Before introducing CC, these plots had average topsoil pH of 6.31, soil organic C of 20.08 g kg⁻¹, bulk density of 1.34 Mg m⁻³, $NO_3^-$ level of 7.65 mg kg⁻¹, and $NH_4^+$ level of 6.15 mg kg⁻¹ [14]. A split-plot arrangement of N fertilization rates (0, 202, and 269; kg N ha⁻¹) and CC (cover crop, CC; bare fallow control, BF) in a randomized complete block design with three replicates was used on the continuous corn production plots. The main plots were 18 m long by 6 m wide, and the subplots were 18 m long and 3 m wide. Corn was planted on 3 June 2019 and 26 May 2020 at 88,000 seeds ha⁻¹. Nitrogen fertilization occurred in early to mid-May with urea ammonium nitrate solution (UAN 28%). No P or K fertilizers or lime were applied. Fertilizer, herbicide, and pest management decisions followed the best management practices for the site as recommended by the Illinois Agronomy Handbook [36]. Cash crop harvesting occurred in mid-October with a plot combine (Almaco, Nevada, IA, USA). Following harvest each year, a CC mixture of cereal rye [*Secale cereale* L.] and hairy vetch [*Vicia villosa* Roth.] was no-till drill-seeded at the rate of 84 kg seeds ha⁻¹ on 3 October 2018 and on 19 October 2019. The CC were terminated in early May after soil sampling and before corn planting with glyphosate [*N*-(phosphonomethyl)glycine] (Roundup WeatherMAX®, Bayer AG, Leverkusen, Germany) at the rate of 1.89 kg a.i. ha⁻¹. Spring tillage was conducted in all plots following CC suppression using a rotary tiller (Dyna Drive Cultivator, EarthMaster, Alamo Group, Inc., Seguin, TX, USA) on 3 June 2019 and on 11 May 2020.



### 2.2. Soil and Cover Crop Biomass Sampling and Determinations

Soil samples were taken on 26 April 2019 and on 30 April 2020. Within each experimental unit, three composited soil subsamples at a depth of 10 cm were taken with an Eijelkamp grass plot sampler (Royal Eijkelkamp Company, Giesbeek, The Netherlands) collecting plugs while walking in a zig–zag pattern. For each plot, a composited soil sample had about 15 plugs, rendering a total of 500 g of soil used for microbial DNA extraction. These soil samples were transported from the experimental site in coolers filled with ice and then stored at −20 °C in the lab. Additionally, three soil core subsamples 0–90 cm in depth per experimental unit were taken using a tractor-mounted soil sampler with soil sleeve inserts (Amity Tech, Fargo, ND, USA). These soil cores were cut at depths of 0–30 cm, 30–60 cm, and 60–90 cm in the lab and composited to determine general soil properties. This study only used the soil property data from the top 0-30 cm soil. Water content was determined by gravimetry (%), and the available soil $NO_3^-$ and ammonium ($NH_4^+$) (mg kg$^{-1}$) were determined using KCl extraction (1:5 ratio of soil to solution) followed by flow injection analysis with a SmartChem 200 (Westco Scientific Instruments, Inc., Danbury, CN). The soil pH (1:1 soil–water) was determined via potentiometry [37] by a commercial laboratory (Brookside Laboratories, Inc., New Bremen, OH, USA). Samples of CC aboveground biomass growth were collected at the same time as soil sampling using three random 0.25 m$^2$ quadrat tosses in each CC plot. The dry weight of the biomass samples was measured after drying them in the oven at 60 °C for 48 h.

### 2.3. Soil DNA Extraction, Sequencing, and Taxonomic Classification

Soil DNA was extracted from 0.25 g of each soil subsample using PowerSoil® DNA isolation kits (MoBio Inc., Carlsbad, CA, USA) following the manufacturer's instructions. The quantity and quality of the extracted DNA were tested using Nanodrop 1000 Spectrophotometer (ThermoFisher Scientific, Waltham, MA, USA), and the extracted DNA samples were stored at −20 °C until analysis. The DNA samples were sequenced for the bacterial V4 region and archaeal 16S rRNA, and the fungal internal transcribed spacer (ITS) region for taxonomic analysis with Illumina MiSeq paired-end System (2 × 250 bp) (Illumina, Inc., San Diego, CA, USA) by the W.M. Keck Center for Comparative and Functional Genomic lab at the University of Illinois Biotechnology Center (Urbana, IL, USA). The sample DNA concentration was limited to 50 ng μL$^{-1}$. The primer sets used for amplification were 515F (GTGYCAGCMGCCGCGGTAA) and 806R (GGAC-TACVSGGGTWTCTAAT) for the bacterial 16S rRNA gene [38], 349F (GTGCAS-CAGKCGMGAAW) and 806R (GGACTACVSGGGTATCTAAT) for the archaeal 16S rRNA gene [39], and 3F (GCATCGATGAAGAACGCAGC) and 4R (TCCTCCGCTTATTGATATGC) for the fungal ITS region [40].

Quality check and processing of the sequences were carried out using QIIME 2.0 pipeline [41,42]. When checking the archaeal demultiplexed sequences from the MiSeq System, not enough sequences were retained for further analysis when the sequences were trimmed at the base positions where the 25th percentile of the quality scores fell below 20 on QIIME 2.0 View [43]. Therefore, the sequences were not trimmed for all bacteria, archaea, and fungi to keep a consistent method across taxa. Nonetheless, bacterial and fungal sequence qualities were mostly very high (25th percentile Q > 30) or at least good (25th percentile Q > 20) throughout the base pair positions, so trimming was unnecessary [43]. The plugin DADA2 was used to denoise and remove chimeric and low-quality sequences with the option chimera-method consensus, and then resulting sequences were clustered into amplicon sequence variants (ASVs) [44]. The rarefaction curve for each bacterial, archaeal, and fungal ASVs plateaued around 900, 150, and 300, respectively, in sampled sequences on average (Figure S2).

The ASVs were classified with Ribosomal Database Project (RDP) web classifier or RDPTool package [45] using 16S rRNA training set 18 and Warcup Fungal ITS trainset 2. RDP database was chosen because it has a lower annotation error rate than other 16S

rRNA databases [46]. The resulting classified ASVs were grouped by genus and those with low (<0.1%) per-sample relative abundances averaged across all samples were filtered out using package dplyr version 1.0.5 [47] in R, version 4.1.0 [48]. The ASV sequences were aligned using MAFFT method [49], and then the maximum-likelihood phylogenetic tree was built using fasttree and midpoint-root methods in QIIME2. The resulting phylogenetic tree was used to calculate the UniFrac distance for β-diversity in QIIME2. The $\alpha$-diversity measures extracted were observed for a number of ASVs for richness, Pielou's evenness parameter, J, for evenness, and Shannon's H′ for diversity. The β-diversity between treatment levels was analyzed with pairwise permutational multivariate analysis of variance (PERMANOVA) that reports the pseudo-F-value for testing the null hypothesis and the probability values before (*p*-value) and after (*q*-value) using the Benjamini–Hochberg false discovery rate (FDR) correction for multiple testing [33,50,51].

### 2.4. Indicator Microbes and Statistical Analysis

After classifying the ASVs to genus-level, a bootstrap forest partitioning method deployed within the JMP® predictor screening platform was used to select the genera sensitive to N fertilization and CC treatments [52–54]. The sparsity in the abundance data of these selected indicator genera was resolved by using zero-replacement with the function cmultRepl from R package zCompositions [55]. These datasets were then normalized by central log-ratio transformation to manage the compositionality of the data [56]. Then, a principal component analysis (PCA) was used as a data reduction technique to further select the indicator genera. Procedure FACTOR in SAS software version 9.4 (SAS Institute, Cary, NC) with priors = 1 summarized the abundances of each genus into a set of uncorrelated composite variables, or principal components (PCs). The PCs with eigenvalues ≥ 1 that also explained at least 5% of the variability in the dataset were used as response variables for statistical analysis. Genera with an important correlation with each PC (loading value > |0.5|) were considered as the bioindicators [57]. Each indicator genus was then searched in the List of Prokaryotic names with Standing in Nomenclature (LPSN), or other primary research, for its known characteristics [58].

Linear mixed models were fitted using the GLIMMIX procedure in SAS software to determine the effects of N rates, CC, and their interactions on soil properties, CC biomass, $\alpha$-diversity measures, and PC scores of the indicator genera [59]. N rates, CC, and their interaction were considered fixed effects, whereas blocks, years, and their interactions with the fixed effects were considered random terms in the analysis of variance (ANOVA). For any significant treatment effects on the response variables based on ANOVA results, the least square means of the response variables were separated by treatment levels, using the lines option and setting the probability of a type I error at $\alpha = 0.1$. The ggplot2 package in R was used to create figures [60]. The figures for indicator genera visualized the combined results of the PCA and mean separation procedure, illustrating the responses of each indicator genus's abundance to N rates, CC, and their interactions. These responses were calculated as the mean PC score for a given treatment level multiplied by the PC loading score of the listed indicator genus, which will be referred to as the "M × L" [33]. To assess the relationships between indicator microbes and the soil properties, R function cor with option method = "pearson" was used to calculate the Pearson's correlation coefficients among the selected soil properties, and the bacterial, archaeal, and fungal PC scores. Here we considered the associations with coefficients above |0.8| as "very strong", those with values between |0.6–0.8| as "strong", and those with values between |0.4–0.6| as "moderate", modified from the ranges used in Huang et al. [21]. The statistical significance of these associations was calculated with rcorr function in R package Hmisc [61] setting the Type I error rate ($\alpha$) at 0.05.

## 3. Results

### 3.1. Soil Properties and Cover Crop Biomass

Table 1 summarizes the estimated treatment means, the standard errors of the mean (SEM), sample size (n), degrees of freedom (df), and the probability values associated with the ANOVA for each source of variation (*p*-values), as well as the results of mean separation procedures for CC biomass and selected soil properties of $NH_4^+$, $NO_3^-$, and soil pH. In this study, Nrate main effect was statistically significant for soil pH ($p = 0.0025$) and CC biomass ($p = 0.0099$). Soil pH decreased sequentially with higher N rates. The CC biomass significantly decreased in the unfertilized controls (N0, 1.87 Mg ha$^{-1}$) when compared with the biomass recorded under both N rates, N202 and N269 (3.20 and 3.29 Mg ha$^{-1}$, respectively). The soil $NO_3^-$ level showed a marginal CC main effect ($p = 0.1601$) where it nearly halved with CC, compared to BF.

**Table 1.** The estimated treatment means, standard errors of the treatment mean (SEM), and the sample size (*n*) of the selected soil chemical properties, including soil ammonium level ($NH_4^+$; mg kg$^{-1}$), nitrate level ($NO_3^-$; mg kg$^{-1}$), and soil pH, and cover crop biomass dry weight (CC biomass; Mg ha$^{-1}$) during the two years of experiment determined by the main effects of N fertilization (Nrate), cover cropping (CC), and the interaction (Nrate × CC). Probability values (*p*-values) and degrees of freedom (df) associated with the different sources of variations from Type III Test analysis of variance results are shown below.

| Treatments | | NH$_4^+$ | | NO$_3^-$ | | pH | | | CC Biomass | | |
|---|---|---|---|---|---|---|---|---|---|---|---|
| | *n* | Mean | SEM | Mean | SEM | Mean | | SEM | *n* | Mean | SEM |
| Nrate [2] | | | | | | | | | | | |
| 0 | 12 | 24.32 | 4.602 | 1.23 | 0.380 | 6.72 | a [1] | 0.126 | 6 | 1.87 | b | 0.273 |
| 202 | 12 | 28.24 | | 1.68 | | 5.82 | b | | 6 | 3.20 | a | |
| 269 | 12 | 25.48 | | 1.76 | | 5.43 | c | | 6 | 3.29 | a | |
| CC [3] | | | | | | | | | | | |
| BF | 18 | 25.20 | 3.654 | 2.06 | 0.426 | 5.92 | | 0.115 | | | |
| CC | 18 | 26.83 | | 1.06 | | 6.06 | | | | | |
| Nrate × CC | | | | | | | | | | | |
| BF0 | 6 | 25.02 | 5.205 | 1.72 | 0.464 | 6.72 | | 0.160 | | | |
| CC0 | 6 | 23.62 | | 0.75 | | 6.72 | | | | | |
| BF202 | 6 | 28.32 | | 2.07 | | 5.72 | | | | | |
| CC202 | 6 | 28.17 | | 1.30 | | 5.92 | | | | | |
| BF269 | 6 | 22.27 | | 2.40 | | 5.32 | | | | | |
| CC269 | 6 | 28.70 | | 1.12 | | 5.53 | | | | | |
| Sources of Variation | df | NH$_4^+$ | | NO$_3^-$ | | pH | | | CC biomass | | |
| Nrate | 2 | 0.7995 | | 0.3440 | | 0.0025 | | | 0.0099 | | |
| CC | 1 | 0.7071 | | 0.1601 | | 0.3079 | | | | | |
| Nrate × CC | 2 | 0.4137 | | 0.2141 | | 0.6806 | | | | | |

[1] The treatment means followed by the same lowercase letter were not statistically different within a given column of each taxon ($\alpha = 0.10$); [2] Nrate levels: 0, 202, and 269 kg N ha$^{-1}$; [3] CC levels: BF, bare fallow control; CC, hairy vetch and cereal rye mixture cover cropping.

### 3.2. Overall Characterization of the Soil Microbial Community

The Supplementary Table S1 summarizes the estimated treatment means, standard errors of the mean (SEM), and the probability values (*p*-values) of the $\alpha$-diversity parameters of bacterial, archaeal, and fungal communities. Supplementary Table S2 shows the PERMANOVA results for the $\beta$-diversity among the treatment levels of Nrate and CC, and their interactions for the three taxa, including the pseudo-F-values and the probability values after correction for multiple comparisons (q-value). For $\alpha$-diversity, the evenness

parameter Pielou's J of bacteria and the observed number of fungal ASVs both showed a statistically significant main effect of Nrate ($p = 0.0279$, and $p = 0.0394$, respectively) (Table S1). The bacterial community became more "even" under N269 compared to N0 and N202. Meanwhile, the number of fungal ASVs decreased more in the unfertilized controls than in the fertilized plots. As for the β-diversity, the differences between bacterial community composition were statistically significant between N0 and N202 (q = 0.0015) and N269 (q = 0.0015), while they were only marginal (q = 0.0740) between N202 and N269 (Table S2). The bacteria β-diversity did not have a statistically significant CC effect (q = 0.2860). The β-diversity between the archaeal community was also statistically significant (q = 0.0030) between N0 and N202 and N269, but did not differ between N202 and N269 (q = 0.2510). Additionally, the archaeal community did not have a significant CC main effect (q = 0.5480). The fungal community structure, on the other hand, showed statistically significant interaction effects of Nrate × CC, as well as CC main effects (q = 0.0010). Thus, the fungal community structure differed between CC and BF across Nrate comparisons. The fungal community structure differed between N0 and N202 and N269 within CC, but not with in BF.

Metabarcoding analysis comprised 2,625,449 bacterial, 199,685 archaeal, and 424,800 fungal sequences. After denoising and removing chimeric sequences, the bacterial sequences were grouped into 778 genera, of which 176 had average relative abundances greater than 0.1%. Likewise, the archaeal sequences were grouped into four genera whose average relative abundances were greater than 0.1%. Lastly, the fungal sequences were grouped into 321 genera, with 144 of them being above average relative abundances of 0.1%. The most abundant bacterial phylum among all classified ASVs was Proteobacteria (34.2%), followed by Actinobacteria (12.0%), Acidobacteria (11.9%), Bacteroidetes (11.2%), and Chloroflexi (5.2%). Archaea was dominated by Thaumarchaeota (90.3%). The most abundant fungal phylum was Ascomycota (59.5%), followed by Basidiomycota (29.7%), Zygomycota (5.7%), and Glomeromycota (3.1%).

### 3.3. Indicators of Cover Crop and N Rate Treatments

Supplementary Tables S3–S5 each summarizes, for bacteria, archaea, and fungi respectively, the eigenvalues and cumulative proportions of the variability in the data set explained by each principal component (PC) in the PCA among the genera selected by predictor screening, along with their respective eigenvectors. The eigenvectors consist of the loading scores on these PCs from each of these selected genera. Tables 2 and 3 each show the estimated treatment means, standard errors of the mean (SEM), their probability values (*p*-values), and the results of the mean separation procedures for bacterial and archaeal PC scores and fungal PC scores, respectively.

**Table 2.** The estimated treatment means and standard errors of the mean (SEM) of each group of principal components (PC) calculated for bacterial and archaeal indicators determined by the main effects of N fertilization (Nrate), cover cropping (CC), and their interaction. Probability values (*p*-values), sample size (*n*), and degrees of freedom (df) associated with the different sources of variation from Type III Test analysis of variance results are shown below.

| Treatment | | Bacteria | | | | | | | Archaea | | |
|---|---|---|---|---|---|---|---|---|---|---|---|
| | | PC1 | | PC2 | | PC3 | PC4 | PC5 | PC1 | | PC2 |
| Nrate [2] | | | | | | | | | | | |
| 0 | | 1.26 | a [1] | −0.25 | | 0.05 | 0.00 | −0.17 | −0.66 | b | −0.10 |
| 202 | | −0.43 | b | 0.40 | | 0.36 | 0.07 | 0.48 | 0.17 | a | 0.16 |
| 269 | | −0.83 | c | −0.14 | | −0.42 | −0.07 | −0.31 | 0.49 | a | −0.06 |
| | SEM | 0.180 | | 0.359 | | 0.687 | 0.492 | 0.517 | 0.323 | | 0.365 |
| CC [3] | | | | | | | | | | | |
| BF | | −0.02 | | −0.60 | b | −0.13 | 0.06 | −0.14 | 0.18 | | 0.15 |
| CC | | 0.02 | | 0.60 | a | 0.13 | −0.06 | 0.14 | −0.18 | | −0.15 |

| | | | Bacteria | | | | | Archaea | |
|---|---|---|---|---|---|---|---|---|---|
| | SEM | | 0.183 | 0.311 | 0.664 | 0.483 | 0.374 | 0.267 | 0.324 |
| Nrate × CC | | | | | | | | | |
| BF0 | | | 1.26 | −0.61 | −0.35 | −0.18 b | −0.63 b | −0.60 | 0.20 |
| CC0 | | | 1.27 | 0.10 | 0.46 | 0.18 ab | 0.28 a | −0.72 | −0.40 |
| BF202 | | | −0.39 | −0.45 | 0.60 | 0.66 a | 0.76 ab | 0.50 | 0.12 |
| CC202 | | | −0.47 | 1.24 | 0.12 | −0.52 b | 0.21 ab | −0.16 | 0.21 |
| BF269 | | | −0.93 | −0.75 | −0.63 | −0.31 b | −0.54 ab | 0.64 | 0.14 |
| CC269 | | | −0.73 | 0.46 | −0.20 | 0.17 ab | −0.07 ab | 0.34 | −0.26 |
| | SEM | | 0.245 | 0.446 | 0.714 | 0.547 | 0.561 | 0.412 | 0.466 |
| Sources of Variation | *n* | df | PC1 | PC2 | PC3 | PC4 | PC5 | PC1 | PC2 |
| Nrate | 12 | 2 | <0.0001 | 0.4017 | 0.1589 | 0.9115 | 0.5350 | 0.0679 | 0.7921 |
| CC | 18 | 1 | 0.8727 | 0.0475 | 0.3274 | 0.7735 | 0.3948 | 0.4031 | 0.3722 |
| Nrate × CC | 6 | 2 | 0.7702 | 0.3717 | 0.1686 | 0.0436 | 0.0415 | 0.7121 | 0.6910 |

[1] The treatment means followed by the same lowercase letter are not statistically different within a given column of each taxon ($\alpha$ = 0.10); [2] Nrate levels: 0, 202, and 269 kg N ha$^{-1}$; [3] CC levels: BF, bare fallow control; CC, hairy vetch and cereal rye mixture cover cropping.

**Table 3.** The estimated treatment means and standard errors of the mean (SEM) of each group of principal components (PC) calculated for fungal indicators determined by the main effects of N fertilization (Nrate), cover cropping (CC), and their interaction. Probability values (*p*-values), sample size (*n*), and degrees of freedom (df) associated with the different sources of variation from Type III Test analysis of variance results are shown below.

| | | **Fungi** | | | | | | |
|---|---|---|---|---|---|---|---|---|
| **Effect** | | **PC1** | **PC2** | **PC3** | **PC4** | **PC5** | **PC6** | **PC7** |
| Nrate [2] | | | | | | | | |
| 0 | | −0.31 | −0.89b [1] | 0.25 | −0.13 | 0.34 | −0.18 | −0.20 |
| 202 | | 0.22 | 0.32a | −0.35 | −0.14 | 0.02 | −0.12 | 0.01 |
| 269 | | 0.09 | 0.56a | 0.10 | 0.27 | −0.36 | 0.29 | 0.18 |
| | SEM | 0.735 | 0.414 | 0.429 | 0.407 | 0.403 | 0.329 | 0.298 |
| CC [3] | | | | | | | | |
| BF | | 0.20 | 0.19 | 0.48 a | 0.27 | 0.32 | 0.23 | 0.15 |
| CC | | −0.20 | −0.19 | −0.48 b | −0.27 | −0.32 | −0.23 | −0.15 |
| | SEM | 0.706 | 0.433 | 0.342 | 0.373 | 0.335 | 0.314 | 0.244 |
| Nrate × CC | | | | | | | | |
| BF0 | | −0.22 | −0.79cd | 0.81 | 0.06 | 0.28 | −0.35bc | −0.10 |
| CC0 | | −0.40 | −0.98d | −0.32 | −0.33 | 0.41 | −0.01ab | −0.30 |
| BF202 | | 0.47 | 0.22b | 0.06 | 0.34 | 0.48 | 0.64a | 0.42 |
| CC202 | | −0.03 | 0.43ab | −0.76 | −0.61 | −0.45 | −0.87c | −0.39 |
| BF269 | | 0.34 | 1.15a | 0.57 | 0.41 | 0.21 | 0.40ab | 0.14 |
| CC269 | | −0.16 | −0.03abc | −0.36 | 0.13 | −0.93 | 0.19ab | 0.23 |
| | SEM | 0.789 | 0.471 | 0.502 | 0.501 | 0.520 | 0.422 | 0.422 |

| Sources of Variation | *n* | df | PC1 | PC2 | PC3 | PC4 | PC5 | PC6 | PC7 |
|---|---|---|---|---|---|---|---|---|---|
| Nrate | 12 | 2 | 0.5901 | 0.0010 | 0.5654 | 0.5473 | 0.4633 | 0.4289 | 0.6725 |
| CC | 18 | 1 | 0.3361 | 0.4656 | 0.0753 | 0.2566 | 0.1782 | 0.3705 | 0.3828 |
| Nrate × CC | 6 | 2 | 0.9079 | 0.0318 | 0.8742 | 0.6983 | 0.3638 | 0.0353 | 0.5602 |

[1] The treatment means followed by the same lowercase letter are not statistically different within a given column of each taxon ($\alpha$ = 0.10); [2] Nrate levels: 0, 202, and 269 kg N ha$^{-1}$; [3] CC levels: BF, bare fallow control; CC, hairy vetch and cereal rye mixture cover cropping.

### 3.3.1. Bacterial and Archaeal Community

For bacteria, five PCs explained 51.6% of the variability in the 68 selected top contributing ASVs. PC1 explained 24.5% of the variability, including positive loadings from genera *Archangium, Arenimonas, Formivibrio, Niabella, Pseudonocardia, Longimicrobium, Thermanaerothrix, Rhodoplanes, Nitrospira, Phaselicystis, Methyloligella, Basilea, Povalibacter,* and uncultured Acidobacteria subgroups 4 and 7; it had negative loadings from *Micropepsis, Porphyrobacter, Baekduia, Nitrobacter, Rhizomicrobium, Chujaibacter, Denitratisoma, Pseudolabrys, Vicingus, Flavitalea,* and uncultured Acidobacteria subgroups 1 and 3 (Table S3). PC1 had a statistically significant N rate main effect ($p < 0.0001$) where its mean PC scores decreased sequentially from N0 to N269, and their differences were statistically significant (Table 2). Therefore, indicator bacteria with positive loadings on PC1 increased in abundance under N0, while those with negative loadings increased with N202 and N269 (Figure 1).

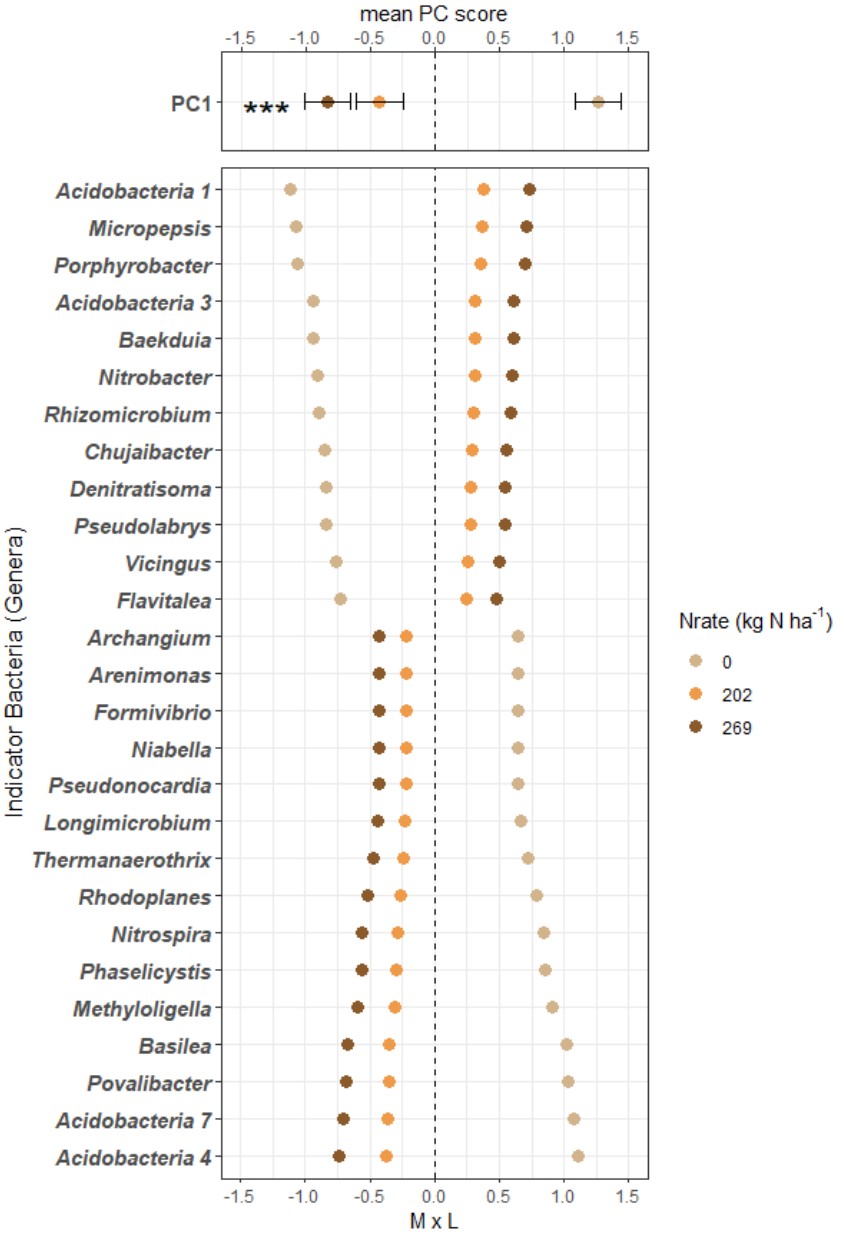

**Figure 1.** The top panel shows the estimated mean principal component (PC) scores of the bacterial PC1 for each level of N rate treatment with their standard errors as whiskers. The asterisks indicate the probability value of the treatment effect from analysis of variance (***: $p < 0.001$). The bottom

panel shows the contribution of each bacterial indicator (genera) to PC1 multiplied by the mean PC scores of each level of N treatment (M × L). The treatment levels are: 0 kg N ha⁻¹ (tan), 202 kg N ha⁻¹ (orange), and 269 kg N ha⁻¹ (brown).

PC2 accounted for 9.7% of the variability and included positive loadings from genera *Panacibacter*, *Racemicystis*, *Mesorhizobium*, *Luteimonas*, *Hydrobacter*, and *Stenotrophobacter*, and negative loadings from *Gemmata* and *Gemmatirosa*. PC2 had a statistically significant ($p = 0.0475$) CC main effect, separating the mean PC scores between BF and CC. The abundances of indicator bacteria that had positive loadings on PC2 thus increased in abundance under CC, while those with negative loadings increased under BF (Figure 2).

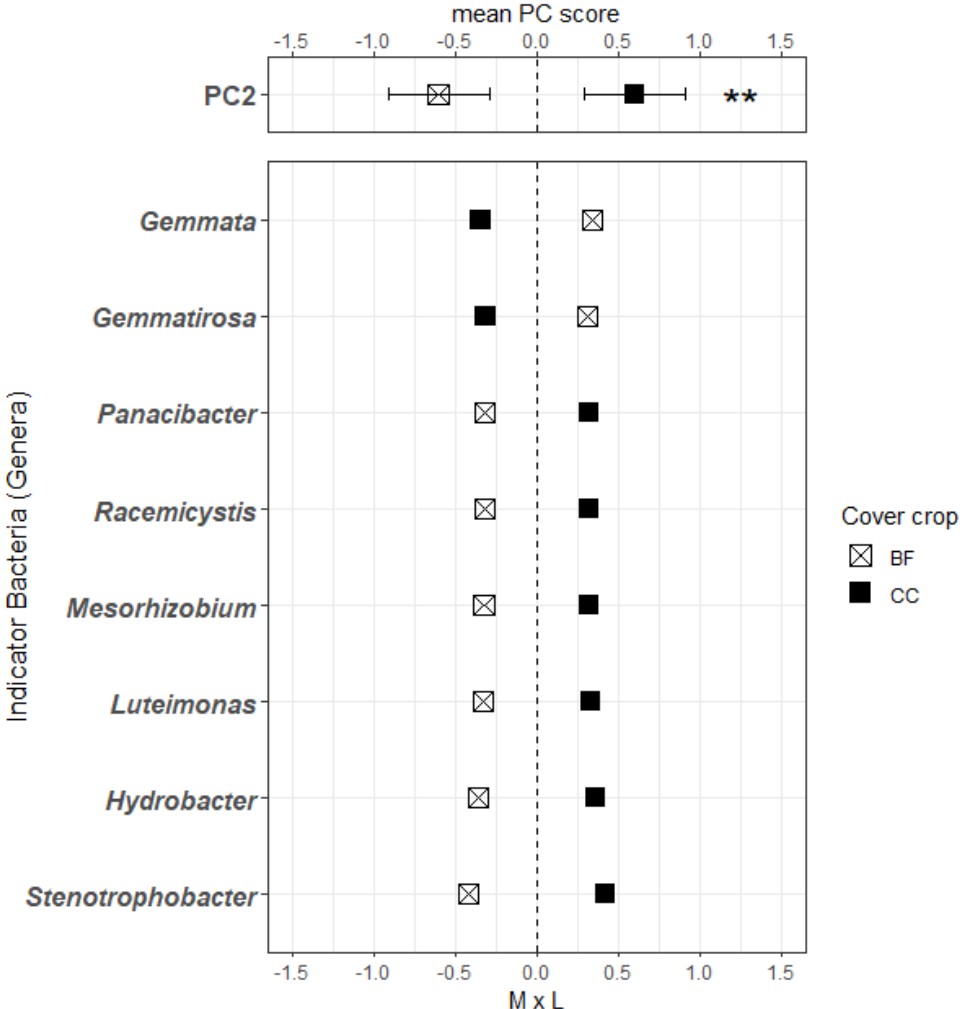

**Figure 2.** The top panel shows the estimated mean principal component (PC) scores of the bacterial PC2 for each level of cover cropping treatment with their standard errors as whiskers. The asterisks indicate the probability value of the treatment effect from analysis of variance (**: $p < 0.05$). The bottom panel shows the contribution of each bacterial indicator to PC2 multiplied by the mean PC scores of each level of cover cropping treatment (M × L). The treatment levels are bare fallow (BF; crossed box) and cover crop mixture (CC; filled box).

PC3 explained 6.4% of the variability and had positive loadings from genera *Caenibius* and negative loadings from uncultured Acidobacteria subgroup 6 and *Nitrolancea*

(Table S3). However, PC3 had no significant effect from the treatments (Table 2). PC4 explained 5.7% of the variability and had a negative loading from genus *Parafilimonas*. PC5 accounted for 5.4% of the variability and had positive loadings from genus *Pirellula* and Bacillariophyta and a negative loading from *Luteimonas*. However, Bacillariophyta is a misnomer of unclassified sequences in the RDP database; thus, this taxon was excluded from further results [62]. Interaction effects were statistically significant for both bacterial PC4 ($p$ = 0.0436) and PC5 ($p$ = 0.0415). For PC4, the mean PC score increased statistically significantly with N202 compared to N0 and N269 within BF; within CC, it rather decreased with N202, but the difference was not statistically significant (Table 2). Therefore, the abundance of *Parafilimonas* decreased within BF202 compared to BF0 and BF269. Under CC, this trend flipped so that its abundance was the highest within CC202, but the differences among N rates were not significant (Figure 3). A statistically significant difference in the mean PC5 scores between CC0 and BF0 was detected (Table 2). Thus, indicator bacteria with positive loadings on PC5 were more abundant with CC0 than BF0, while those with negative loadings increased in abundance with BF0 than CC0 (Figure 4).

The archaeal PC1 explained 48.3% of the variability in the archaeal data and included negative loading from the genus *Nitrososphaera* and positive loadings from the uncultured Woesearchaeota AR16 and AR20 (Table S4). The mean scores of PC1 had a marginally significant statistical main effect of Nrate ($p$ = 0.0679) where they were greater with N202 and N269 than N0 (Table 2). Therefore, with higher N rates, *Nitrososphaera* decreased in abundance while AR16 and AR20 increased (Figure 5). Archaeal PC2 explained 29.2% of the variability in the data and included positive loading from the genus *Methanomassiliicoccus*. However, PC2 did not have any significant effects from the treatments (Table 2).

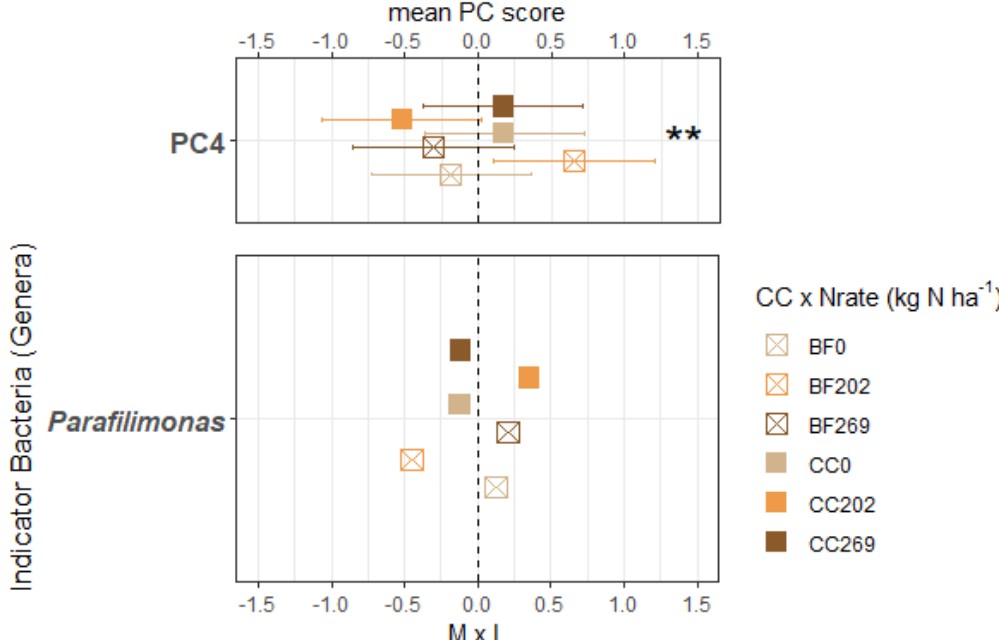

**Figure 3.** The top panel shows the estimated mean principal component (PC) scores of the bacterial PC4 for each level of N rate and cover cropping (CC × Nrate) treatment interactions with their standard errors as whiskers. The asterisks indicate the probability value of the treatment effect from analysis of variance (**: $p$ < 0.05). The bottom panel shows the contribution of each bacterial indicator to PC4 multiplied by the mean PC scores of each level of N rate and cover cropping treatment interactions (M × L). The N rate treatment levels are: 0 kg N ha⁻¹ (tan), 202 kg N ha⁻¹ (orange), and 269 kg N ha⁻¹ (brown). The cover cropping treatment levels are bare fallow (BF; crossed box) and cover crop mixture (CC; filled box).

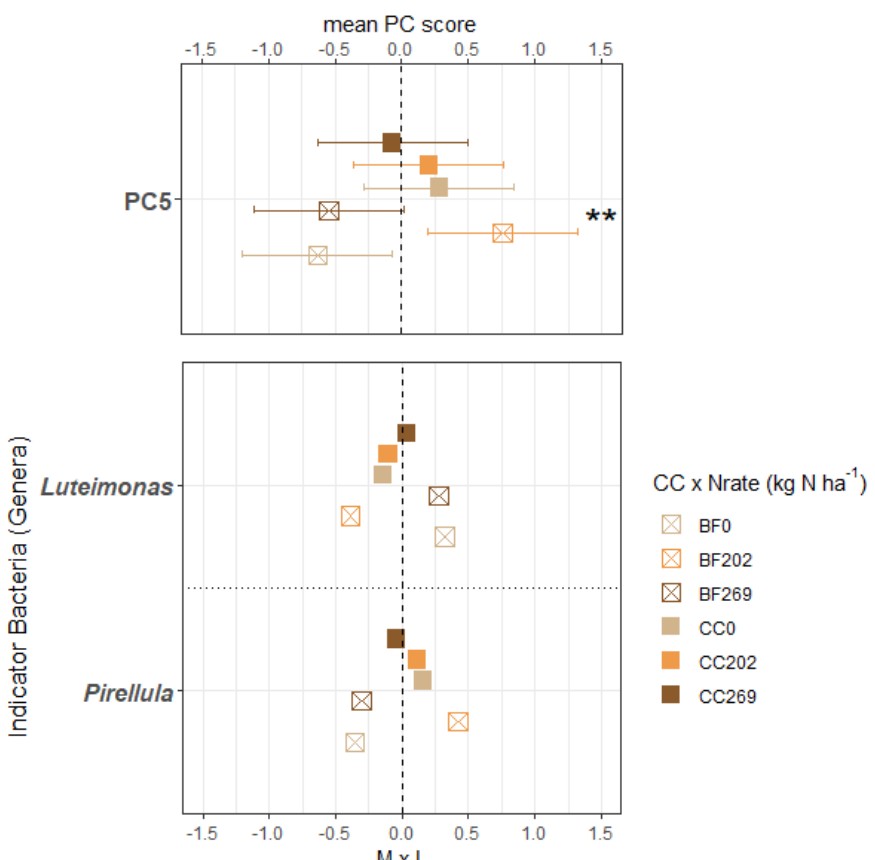

**Figure 4.** The top panel shows the estimated mean principal component (PC) scores of the bacterial PC5 for each level of N rate and cover cropping (CC × Nrate) treatment interaction with their standard errors of the mean as whiskers. The asterisks indicate the probability value of the treatment effect from analysis of variance (**: $p < 0.05$). The bottom panel shows the contribution of each bacterial indicator to PC5 multiplied by the mean PC scores of each level of N rate and cover cropping treatment interactions (M × L). The N rate treatment levels are: 0 kg N ha$^{-1}$ (tan), 202 kg N ha$^{-1}$ (orange), and 269 kg N ha$^{-1}$ (brown). The cover cropping treatment levels are bare fallow (BF; crossed box) and cover crop mixture (CC; filled box).

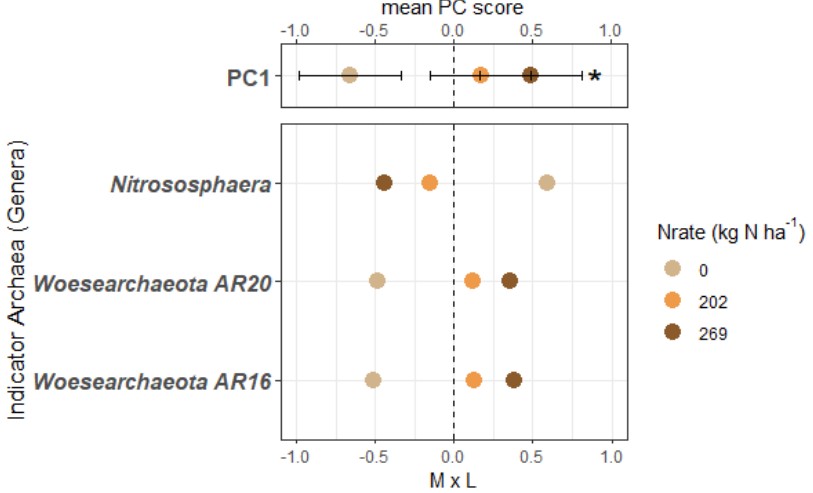

**Figure 5.** The top panel shows the estimated mean principal component (PC) scores of the archaeal PC1 for each level of N rate treatment with their standard errors of the mean as whiskers. The asterisks indicate the probability value of the treatment effect from analysis of variance (*: $p < 0.1$). The

bottom panel shows the contribution of each archaeal bioindicator to PC1 multiplied by the mean PC scores of each level of N treatment (M × L). The treatment levels are: 0 kg N ha⁻¹ (tan), 202 kg N ha⁻¹ (orange), and 269 kg N ha⁻¹ (brown).

### 3.3.2. Fungal Community

Seven PCs explained a total of 58.8% of the variability among 36 selected top contributing ASVs (Table S5). PC1 explained 16.0% of the variability and included positive loadings from genera *Acremonium*, *Alternaria*, *Davidiella*, *Exophiala*, *Phaeosphaeria*, and *Phaeosphaeriopsis* and negative loadings from *Coemansia*, *Glomus*, and *Mortierella*. However, fungal PC1 did not have a statistically significant treatment effect (Table 3). PC2 accounted for 11.3% of the variability in the data and included positive loadings from genera *Podospora*, *Sporobolomyces*, and *Pestalotiopsis* and negative loadings from *Tetracladium*, *Ajellomyces*, and *Edenia*. PC2 had a statistically significant ($p = 0.0318$) Nrate × CC interaction effect. The mean PC2 scores generally increased with N fertilization for both CC and BF; within BF, the mean PC2 score increased sequentially with higher N rates, but it was rather smaller with N269 than N202 within CC (Figure 6). The mean separation results in Table 3 also show a statistically significant difference between BF202 and BF269. Therefore, the abundances of indicator fungi with positive loadings on PC2 generally increased with N fertilization, so that they continued to increase with BF269 but not with CC269. Conversely, those with negative loadings on PC2 showed the opposite trend where their abundances decreased with N fertilization.

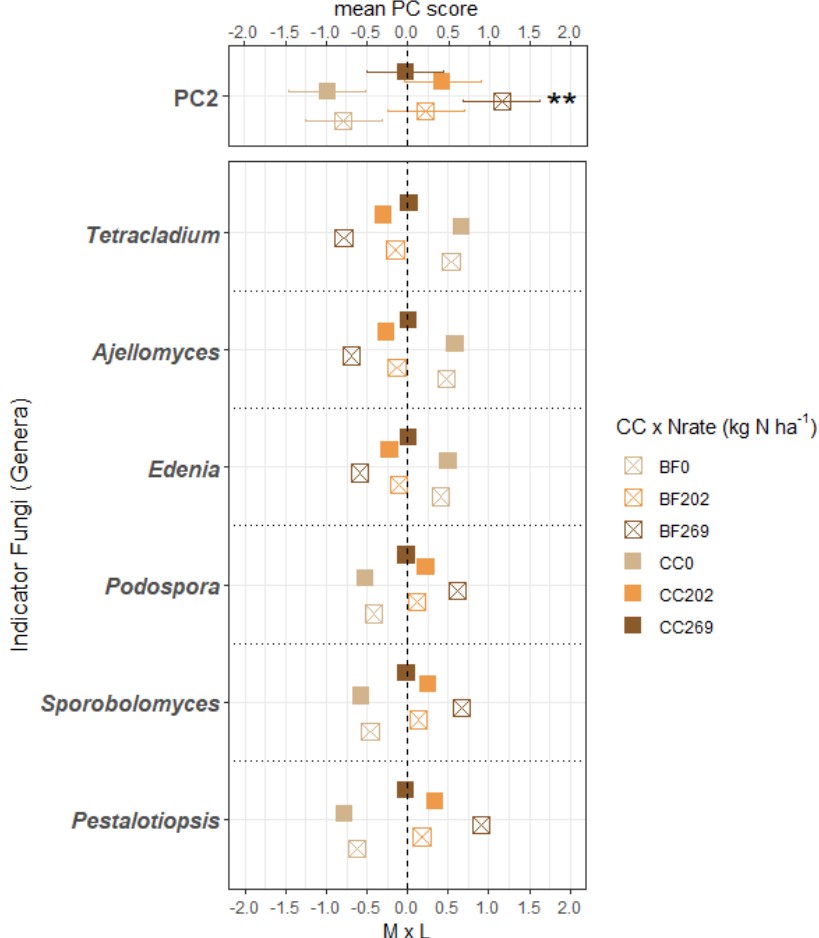

**Figure 6.** The top panel shows the estimated mean principal component (PC) scores of the fungal PC2 for each level of N rate and cover cropping (CC × Nrate) treatment interaction with their standard errors of the mean as whiskers. The asterisks indicate the probability value of the treatment

effect from analysis of variance (\*\*: $p < 0.05$). The bottom panel shows the contribution of each fungal indicator to PC2 multiplied by the mean PC scores of each level of N rate and cover cropping treatment interaction (M × L). The N rate treatment levels are: 0 kg N ha$^{-1}$ (tan), 202 kg N ha$^{-1}$ (orange), and 269 kg N ha$^{-1}$ (brown). The cover cropping treatment levels are bare fallow (BF; crossed box) and cover crop mixture (CC; filled box).

PC3 explained 7.3% of the variability and included a positive loading from genus *Talaromyces* and a negative loading from *Albatrellus*. PC3 had a marginally significant statistical effect of ($p = 0.0753$) for the CC main effect, separating the mean PC scores between CC and BF treatment. Thus, *Talaromyces* increased in abundance with BF, while *Albarellus* did so with CC (Figure 7). PC4 accounted for 6.9% of the variability and included positive loadings from genera *Gibberella* and *Phoma*. PC5 explained 6.0% of the variability and included a negative loading from the genus *Guehomyces*. However, PC4 and PC5 did not have statistically significant responses to treatment effects. PC6 explained 5.6% of the variability but did not have ASVs with significant loading scores. PC6 had a statistically significant ($p = 0.0353$) Nrate × CC interaction effect where the mean scores differed statistically significantly between BF202 and BF0 and CC202, with other interactions being intermediate. PC7 accounted for 5.6% of the variability and included a negative loading from the genus *Tetraploa*. PC7 did not have a statistically significant treatment effect.

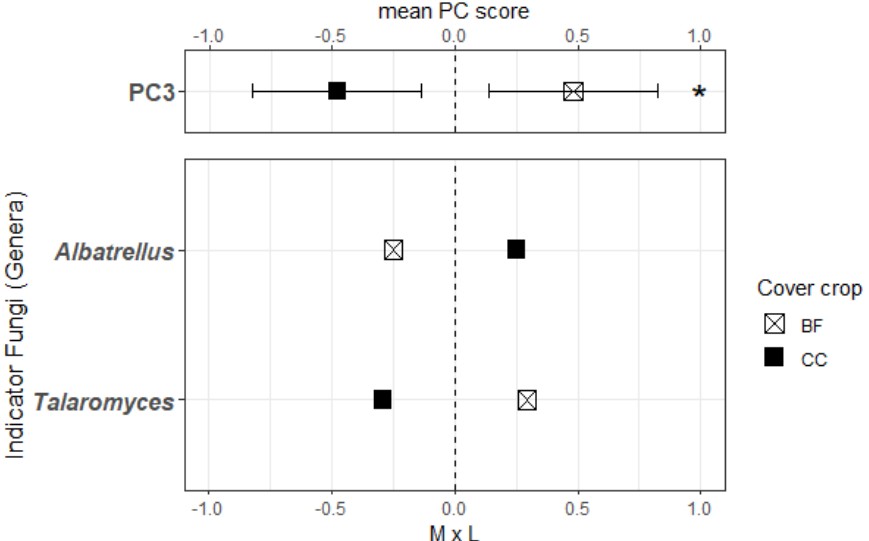

**Figure 7.** The top panel shows the estimated mean principal component (PC) scores of the fungal PC3 for each level of cover cropping treatment with their standard errors of the mean as whiskers. The asterisks indicate the probability value of the treatment effect from analysis of variance (\*: $p < 0.1$). The bottom panel shows the contribution of each fungal indicator to PC3 multiplied by the mean PC scores of each level of cover cropping treatment (M × L). The treatment levels are bare fallow (BF; crossed box) and cover crop mixture (CC; filled box).

### 3.4. Pearson's Correlation Matrix among Variables

A heatmap in Figure 8 visualizes the Pearson's correlation matrix, showing the coefficients among the bacterial, fungal, and archaeal PC scores (BPC# for bacteria, FPC# for fungi, and APC# for archaea), $NH_4^+$, $NO_3^-$, and soil pH. Overall, we found one very strong (>|0.8|), three strong (|0.6–0.8|), and six moderate (|0.4–0.6|) associations of statistical significance ($p < 0.05$). As expected, when following a PCA, bacterial, fungal, and archaeal PCs were not correlated within their respective taxa. BPC1 had very strong positive associations with soil pH. Meanwhile, BPC1 was associated negatively and strongly with FPC2

and moderately with APC1. BPC2 had moderately negative associations with FPC3 and NO$_3^-$. BPC3 did not have any significant association. BPC4 was associated strongly and positively with FPC1. BPC5 did not have significant associations. FPC1 was associated moderately and positively with NO$_3^-$, and negatively with NH$_4^+$. FPC2 was associated negatively and strongly with soil pH. Besides those already described, FPC3, FPC4, FPC5, FPC6, and FPC7 did not have additional significant correlations. In addition to a moderate negative association with BPC1, APC1 also had a moderate negative association with soil pH. Lastly, APC2 did not show any statistically significant association with soil properties or the bacterial or fungal PCs.

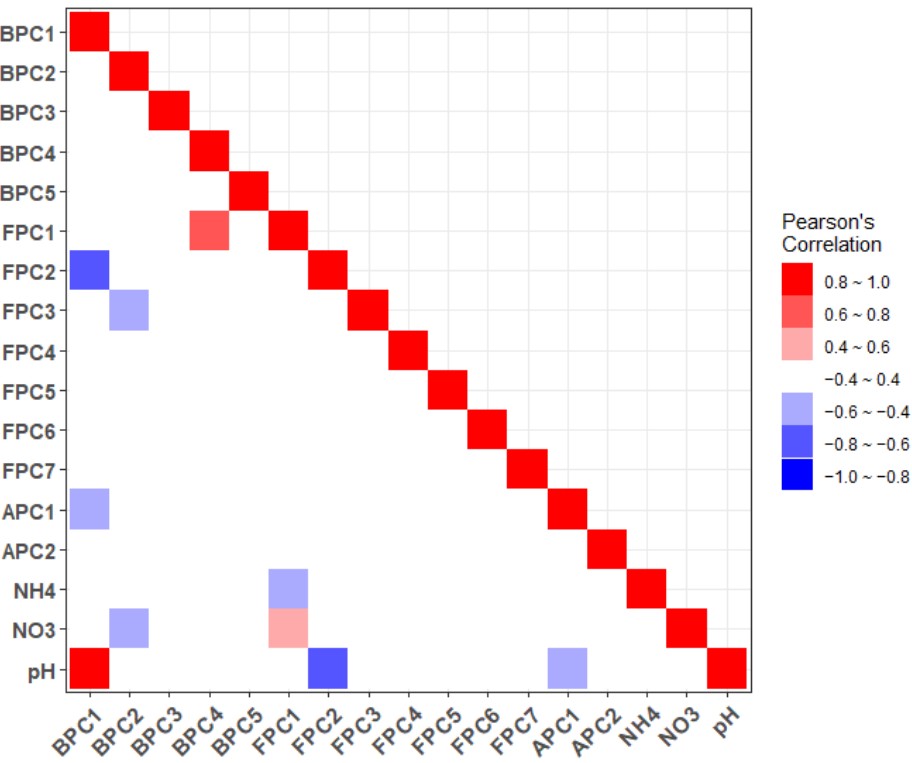

**Figure 8.** The heatmap depicting the matrix of Pearson's correlation coefficients among the principal components (PCs) of the bioindicators and the selected soil properties: soil ammonium (NH4), nitrate (NO3), and pH. The red and blue hues indicate positive and negative associations, respectively. The higher saturation of these colors indicates greater absolute values of Pearson's correlation as shown in the legend on the right. BPC#, bacterial PC; FPC#, fungal PC; APC#, archaeal PC.

## 4. Discussion

### 4.1. Short-Term Agronomic Effects of Legume and Grass Cover Crop Mixture

Overall, this study observed that N fertilization acidified the topsoil by more than a unit in pH. This was expected as protons from nitrification of N fertilizers and increased crop root nutrient uptake, among many other factors, are known to acidify the soil [14,63]. As hypothesized, CC decreased soil NO$_3^-$ levels in this study, likely by scavenging it. Similarly, Acuña and Villamil [64] evaluated the short-term effects of CC on soil properties under soybean production in Illinois and found a significant decrease in soil NO$_3^-$ each spring after the CC season. The NO$_3^-$ reduction by CC also agrees with the significant increase in CC biomass with N fertilization, showing that CC indeed assimilated the excess soil NO$_3^-$ as biomass, decreasing the risk of NO$_3^-$ leaching in this system [3,16]. On the contrary, NH$_4^+$ did not respond significantly to CC. This showed that cereal rye and hairy vetch CC preferred the uptake of NO$_3^-$ over NH$_4^+$ in corn monoculture, suggesting

that CC may not alleviate soil acidification from nitrification. Another study on the effects of tillage, CC, and crop rotation on Missouri Entisols also reported a pH increase of about 0.2 with grass CC within corn monoculture, similar to this study [65]. Overall, our results demonstrate that while CC can effectively reduce the risks of $NO_3^-$ leaching, it has a limited ability to alleviate soil acidification from $NH_4^+$ fertilizers.

### 4.2. Bacterial and Archaeal Indicators

4.2.1. Indicators of N Fertilization

Bacterial genera sensitive to N rate treatment were primarily grouped in PC1. Of its 27 indicator genera, 12 increased in abundance under N fertilization, while 15 experienced decreased abundance. Interestingly, some of the genera favored by N fertilization were also responsive to high N rates within corn monoculture in 2015–2016 data from Villamil et al. [31]: uncultured Acidobacteria subgroup Gp1, *Micropepsis*, *Porphyrobacter*, *Denitratisoma*, *Rhizomicrobium*, *Chujaibacter*, and *Pseudolabrys*. The consistent associations of these indicators with high N rates across studies of the same site suggested that these genera are reliable indicators of soil environment under heavy N fertilization. Here, the N rate main effect does not exclude the underlying CC effect in our model. Thus, the persistent associations of these indicators with N fertilizers across studies imply that the changes brought about by N fertilization in the soil environment have overwhelmed those induced by introducing CC. Indeed, our study and that of Villamil et al. [31] did not share any genera favored by unfertilized control, which suggests the two studies used different microbial communities in unfertilized soils. Therefore, the different assortments of indicators before [31] and after introducing CC (this study) indicated that CC did shift the soil microbial community of corn monocultures. However, this shift was outshined by the dominating effects of N fertilization.

As demonstrated by a very strong association between soil pH and bacterial PC1, soil acidification from N fertilization seems to be a primary factor that shaped the soil environment for the microbes across studies. Villamil et al. [31] observed that the acidophilic indicators flourished with soil acidification from N fertilization. Indeed, some of the indicators favored by N-input have been characterized or suspected to be acidophiles, including *Micropepsis* [66], *Rhizomicrobium* [67], and Acidobacteria subgroup Gp1 and Gp3 [68]. The opposite also held as some of the indicators favored by unfertilized control were either associated with higher pH (Acidobacteria subgroup Gp4 [69]; *Nitrospira* [70]), or were neutrophilic (*Archangium* [71]; *Formivibrio* [72]; *Rhodoplanes* [73]; *Povalibacter* [74]; *Thermanaerothrix* [75]), or alkaliphilic (*Arenimonas* [76]). Soil pH has already been recognized as a primary modulating factor for the bacterial community, as demonstrated by Wu et al. [77], who found bacterial diversity indices decreased at lower pH. Additionally, Ma et al. [78] observed a strong association between soil pH and bacterial β-diversity after 35 years of NPK fertilization in Chinese Mollisols. Therefore, our bioindicators further suggest that the pH-sensitive guilds largely dictate the shifts in the bacterial community upon soil acidification from excessive N inputs.

Many of these N-rate-associated indicators had potential roles in the soil microbial N-cycling. Of those that increased with N fertilization, *Nitrobacter* is a well-known genus of nitrifiers that oxidize nitrite ($NO_2^-$) into $NO_3^-$ [79]. Their proton-producing $NO_2^-$ oxidation could have contributed to soil acidification that favors the acidophilic indicators mentioned above. Meanwhile, *Baekduia* includes denitrifiers that reduce $NO_3^-$ to $NO_2^-$ [80], and *Denitratisoma* includes denitrifiers that reduce $NO_3^-$ to $N_2O$ and nitrogen ($N_2$) gas [81]. The abundance of $NO_3^-$ substrates from fertilizers and nitrification would have promoted these denitrifiers, while the $NO_2^-$ reduced from $NO_3^-$ by *Baekduia* would, in turn, promote the growth and activity of *Nitrobacter*. Likewise, Villamil et al. [31] identified several bioindicators of denitrification that increased with N fertilization (Acidobacteria subgroup Gp1, *Denitratisoma*, *Dokdonella*, and *Thermomonas*). These findings agree well with the meta-analysis of field studies by Ouyang et al. [82] that found consistent increases in the

abundances of nitrifying (*amoA*) and denitrifying (*nirK*, *nirS*, and *nosZ*) genes with N rates over 200 kg N ha$^{-1}$. Therefore, our indicators suggest that frequent N fertilization at above-optimal rates typical for the corn monoculture may augment the nitrifying and denitrifying communities and contribute to the risk of soil N loss as $NO_3^-$ and $N_2O$.

A longer list of N-cycling indicators associated with unfertilized control. *Povalibacter* is known to assimilate N by reducing $NO_3^-$ and $NO_2^-$ into $NH_4^+$ [74]. For the nitrifiers, *Pseudonocardia* includes ammonia-oxidizers and species capable of heterotrophic nitrification [83]. The genus *Niabella* potentially includes heterotrophic nitrifiers as well [84,85]. *Nitrospira* includes known chemolithotrophic $NO_2^-$ oxidizers [86] and complete ammonia oxidizers (comammox) [87]. As for the denitrifiers, *Rhodoplanes* is known for photoorganotrophy, but also completely denitrifies $NO_3^-$ into $N_2$ gas under darkness [73]. *Arenimonas* is a complete denitrifier as well [88]. While *Thermanaerothrix* is known to harbor $NO_2^-$ reducing gene, *nirS*, its denitrification capacity is not yet confirmed [89]. Detecting these indicators involved in diverse N metabolisms can be explained by the inorganic N deficiency being a major ecological pressure in unfertilized soils. Under such pressure, the microbes represented by *Pseudonocardia* and *Niabella* could heterotrophically nitrify organic N, instead of $NH_3$, into $NO_3^-$ [90]. Even if $NH_3$ is partially nitrified, the guild represented by *Nitrospira* could complete the nitrification. Subsequently, these nitrifiers could supply the $NO_3^-$ to the denitrifiers represented by *Rhodoplanes* and *Arenimonas*, which they will completely denitrify into $N_2$ gas. Thus, the two complete denitrifiers (*Rhodoplanes* and *Arenimonas*) and *Povalibacter* (reduces $NO_3^-$ and $NO_2^-$ into $NH_4^+$) imply less risk of $N_2O$ emission and $NO_3^-$ leaching, as the $NO_3^-$ from nitrifiers could be either assimilated back to $NH_4^+$ or be completely denitrified into $N_2$ gas. Although heterotrophic nitrifiers could denitrify and produce $N_2O$, this process is associated with low pH condition, which should be less of the case for unfertilized soils [90]. Therefore, these bioindicators suggested that the microbial N-cycling unaffected by heavy N input may have greater functional diversity and redundancy. Nonetheless, changes in the abundances of these indicators do not warrant subsequent changes in soil N-cycling, because abundance may not translate to activity and not all members of these genera necessarily perform N-cycling. Thus, the results of this study should be complemented by analyses on the overall soil N-cycling and on the microbial functionality, such as enzyme assays and functional genes.

Besides the sensitivity to pH and involvement in the soil N-cycling, bioindicators in PC1 also have been characterized for other interesting properties. Among those that preferred N fertilization, *Micropepsis* [91] and *Rhizomicrobium* [67] have fermentative metabolisms. *Rhizomicrobium* can reduce ferrous and ferric iron in the presence of glucose, which may be an adaptation to iron that readily oxidizes into a ferrous state in acidic soils, and to the overall increase in the iron solubility with decreasing soil pH [67,92]. Indeed, soil iron level increased sequentially with a higher N rate, with 202 kg N ha$^{-1}$ being intermediate, in the topsoil (data not shown). As for those favored by unfertilized control, *Longimicrobium* is a known oligotrophic genus adapted to low nutrient concentration, which is consistent with the relatively nutrient-poor conditions of the control [93]. *Methyloligella* includes specialized obligatory methylotrophs that reduce single carbon compounds as carbon source but does not grow on methane [94]. Understanding what these indicators and their relations with soil properties signify within the microbial network however, will require more exploration. Thus, future efforts should expand our findings, improving the characterization of more soil microbial taxa in field settings and their interconnection with their soil environment.

### 4.2.2. Archaeal Indicators

The archaeal indicators were primarily sensitive to N fertilization, as demonstrated by archaeal PC1. This PC was moderately associated with soil pH, similar to bacterial PC1 as discussed above. Its component, *Nitrososphaera*, increased in abundance with unfertilized control, which is expected considering that this ammonia-oxidizing archaea (AOA) is neutrophilic [95] and oligotrophic [96]. Thus, it may not be well adapted to the relatively

acidic and N-rich environment under heavy N input. This observation also agrees with Yu et al. [97] who studied the relationships between the ammonia-oxidizing community and soil biogeochemical processes, reporting a positive correlation between soil pH and the family Nitrososphaeraceae that includes *Nitrososphaera*. Therefore, our results well demonstrated that *Nitrososphaera* flourishes as neutrophilic and oligotrophic AOA in an unfertilized agroecosystem. This poses the possibility that *Nitrososphaera* contributes to nitrification in this system along with the bacterial nitrifiers described above. Meanwhile, PCA suggested that uncultured Woesearchaeota AR16 increased with N fertilization. While these uncultured genera are much less studied, Woesearchaeota AR16 is found to have some association with soil pH and nitrogen level, consistent with its preference for N fertilization [98]. This study used the universal primer that targets 16S rRNA of both bacteria and archaea because of its common use. The small number of archaeal sequences and indicators compared to those of bacteria and fungi in this study demonstrated that these primers have low specificity to this domain and might not fully capture the archaeal diversity [99]. Therefore, future efforts that focus on archaeal communities using more specific primer sets may reveal more archaeal indicators of this system.

### 4.2.3. Indicators of Cover Cropping

The bacterial indicators that responded to CC were mainly grouped in PC2 (Figure 2). Of the indicators that increased in abundance with CC, *Mesorhizobium* is a known nodule-forming N-fixing symbiont of legumes including the species of *Vicia* [100,101]. The abundance of this genus with legume-grass CC mixture indicated that hairy vetch CC may recruit N-fixers during their growth [102]. Additionally, *Mesorhizobium* includes species that reduce $NO_2^-$ into $N_2O$ under both aerobic and anaerobic conditions [103]. Similarly, *Luteimonas* includes known denitrifiers that reduce $NO_2^-$ into nitric oxide [104], and one of its species *L. memphitis* reduces $NO_2^-$ into $N_2O$ [105]. Additionally, as hypothesized, genera associated with CC indicated diverse niches with unique metabolic or adaptive characteristics. For example, genus *Racemicystis* includes species of various properties including desiccation resistance, bacteria, and yeast lysis, growth under starch, fructose, and glucose, and inability to lyse cellulose [106]. *Panacibacter* grows optimally in near-neutral pH and does not hydrolyze starch and cellulose but grows on other various sugars [107]. *Stenotrophobacter* is a suspected oligotroph [108] and has been observed to increase in abundance with CC [109]. Conversely, only two indicators increased in abundance with bare fallow. The species of *Gemmata* can perform heterotrophic nitrification and anaerobic ammonia oxidation (anammox) [87]. *Gemmatirosa* includes known oligotrophic chemoheterotrophs holding $N_2O$ reducing *nosZ* gene [110,111].

The four N-cycling indicators that responded to CC (*Gemmata*, *Gemmatirosa*, *Luteimonas*, and *Mesorhizobium*) suggest that the guilds that they represent may mediate the fate of $N_2O$ in fertilized CC soil; with CC, less $NO_2^-$ may be directly converted into $N_2$ gas (fewer *Gemmata*), more $NO_2^-$ is reduced to $N_2O$ (more *Luteimonas* and *Mesorhizobium*), but less $N_2O$ is reduced to $N_2$ gas (fewer *Gemmatirosa*). These indicators imply that the relationship between CC and the denitrifier guilds may not be as simple as initially hypothesized. Indeed, a field and lab study in Illinois Mollisols by Foltz et al. [112] reported that grass CC decreased $N_2O$ emissions in the field of corn fertilized at 180 kg N ha$^{-1}$, but that the soil with CC showed greater denitrification potential and $N_2O$ emissions in laboratory assays. The authors explained that the rye CC decreased $N_2O$ emissions by immobilizing its $NO_3^-$ substrate in a field setting, but adding abundant labile C (glucose) and N ($NO_3^-$) under an assay setting led to more emissions in CC soil than the bare control [112]. Meanwhile, a climate chamber study in Germany by Wang et al. [113], on perennial ryegrass [*Lolium perenne* L.] growth, reported more $NO_2^-$-reducing *nirK* genes with ryegrass growth regardless of N rates (0, 50, 100, 200 kg N ha$^{-1}$), although the ryegrass growth still decreased the $N_2O$ emissions by scavenging soil N. The authors explained that the labile C from CC root exudates promoted the *nirK*-holding $NO_2^-$ reducers, regardless of nutrient

availability [113]. Therefore, the observed CC effects in this study on the denitrifying bio-indicators support the scenario that CC modulates the soil nutrient availability via means such as root exudates to make the soil microbial community more prone to $N_2O$ production. Yet, the actual $N_2O$ production is determined by conditions such as net $NO_3^-$ availability, labile C availability from residue decomposition, and waterlogging [20]. Nonetheless, this scenario should be further investigated with studies that encompass functional genes, potential denitrification rates, and $N_2O$ emissions.

Meanwhile, the bacterial indicators in PC4 and PC5 showed a N fertilization and CC interaction effect. The genus *Parafilimonas* in PC4 includes neutrophilic decomposers [114] whose positive response to an intermediate N rate and CC could be a combined result of this genus exploiting the greater residue return from CC and fertilizer input and negatively responding to soil acidification from the highest N rate. As for PC5, besides *Luteimonas*, which is already discussed with PC2, the genus *Pirellula* includes species that perform anammox [87]. Performing ANOVA and mean separation on its gene counts (normalized by central log-ratio) revealed a statistically significant interaction effect ($p =$ 0.0105; data not shown) where their mean was greater in unfertilized CC soil compared to all other N rate and CC combinations (Figure S3). Thus, this genus may work against $N_2O$ emissions by oxidizing $NO_2^-$ and $NH_4^+$ into $N_2$ gas if the soil's N availability is low. Along with the four above-mentioned N-cycling bioindicators of CC, the sensitivity of *Pirellula* to both CC and N input further suggests that soil N availability may play a role in microbially mediated $N_2O$ emission under CC.

In this study, more indicators responded positively to CC than bare fallow, and the indicators of CC displayed various characteristics and functions, compared to only two indicators of bare fallow that were mainly involved in the microbial N-cycling. These results suggest that introducing CC enhanced the soil biodiversity of corn monoculture. Meanwhile, significantly fewer bacterial indicators responded to the CC main effect than that of the N rate. The β-diversity results reflect this since the bacterial community composition did not differ significantly after CC introduction, unlike among the three N rates (Table S2). As speculated earlier with bacterial PC1, perhaps CC has a relatively smaller impact on the bacterial community of corn monoculture compared to the overwhelming changes brought about by decades of annual N fertilization. Nonetheless, the CC impact on the bacterial community should not be underplayed as the indicators imply that this practice may increase the soil biodiversity and affect the soil microbial N-cycling. Since this study had only two years of CC, future efforts with longer-term CC should test whether its effects can accumulate over the years.

### 4.3. Fungal Indicators

The fungal indicators responsive to main treatment effects comprised PC3. The genus *Albatrellus* includes mycorrhizae, but they are known to associate with coniferous hosts [115]. Yet, *Albatrellus* associated very strongly with CC because this genus was nearly absent in bare fallow (only 1.7% of this genus's total gene counts; Figure S3) despite being one of the most abundant fungal genera in the data. This suggests that the members of this genus may have a wider range of hosts or have other unknown relationships with the CC species of this study. As for the genus *Talaromyces*, eight of its species and three Penicillium teleomorphs were observed. This ubiquitous genus occupies a wide range of niches, including endophytes, which promote plant growth and resistance, antagonists to plant pathogens, and those associated with insects [116,117]. Since *Albatrellus* and *Talaromyces* were two of the most abundant genera in the data, their differences in abundance between CC treatments may have driven the significant β-diversity results for the fungal community (Table S2). These results agree well with those of Castle et al. [29], who found winter rye CC to be the stronger factor for fungal community structure than N rates in a corn–soybean rotation in Minnesota.

Meanwhile, the six fungal genera in PC2 responded significantly to the N fertilization and CC interaction effect. Performing ANOVA and mean separation on the gene copy

counts (normalized by central log-ratio) of the genus *Tetracladium* displayed a clear sequential decrease in its abundances with increasing N rates that was statistically significant ($p < 0.0001$; data not shown) (Figure S3). *Tetracladium* includes saprotrophs [118] and endophytes [119] and prefers neutral to slightly acidic soil pH [120]. Therefore, its negative response to higher N rates seems largely driven by soil acidification, which also agrees with fungal PC2 associating negatively with soil pH, as this genus had a negative loading on PC2. A negative loading from Genus *Edenia* to PC2 suggests that this genus responded positively to the unfertilized control, which agrees with the results of Bello et al. [121], who observed a negative correlation between this genus and soil $NO_3^-$ levels in a study on the effects of biochar application on fungal community structure after one year of corn on Chinese Mollisols.

Of the fungal indicators that preferred N fertilization, *Podospora* includes known coprophilous fungi that are associated with animal dung [122] and endophytes of plant barks and shoots [123]. *Sporobolomyces* is a known yeast genus common in agricultural soil and on the phyllosphere of crops, which may explain its positive response to the treatment with the highest N rate, which typically leads to a higher crop yield [124]. Moreover, in a Spanish study by Illescas et al. [125], calcium nitrate fertilization helped this genus to colonize wheat. Thus, our results suggest that this genus might also be associated with corn and benefits from N fertilization. Finally, *Pestalotiopsis* occupies a wide spectrum of niches from plant pathogens, endophytes, and saprobes [123,126]. Thus, this genus may have responded positively to an increased crop yield and residue with N fertilization. Compared to bacterial indicators, fungal genera did not show clear overarching ecological patterns between their known characteristics and their responses to treatments. Nonetheless, the strong association between fungal PC2 and soil pH, and the increased number of observed ASVs with N fertilization (Table S1) suggest that soil pH and soil N availability could have acted as major modulating factors for these fungal indicators. Moreover, considering that many of the fungal genera included endophytes, further research on the crop microbiome might elucidate the potential endophytic relationship among the fungal indicators of this study, crop production, and the soil fungal community.

## 5. Conclusions

This study provided a unique opportunity to use bioindicators to characterize and monitor the soil microbial community upon introducing cover crops to an intensely managed corn monoculture common in the US Midwest region. In this study, most of the bioindicators had known characteristics that could reasonably explain their responses to the treatments. Therefore, we found that genus-level indicators with high-taxonomic resolution can provide detailed insights into the soil microbiota that were inaccessible for past taxonomic and functional indicators. Namely, the opposite responses to N fertilization from acidophilic bioindicators versus those of neutrophiles and alkaliphiles demonstrated that soil acidification from N fertilizers dominated the soil microbiota of this system. The N-cycling bioindicators suggested that N fertilization may stimulate the nitrifiers and denitrifiers. Conversely, unfertilized soils may form a more diverse N-cycling community with functions that might mitigate the risks of $NO_3^-$ leaching and $N_2O$ emissions. The bioindicators under cover cropping indicated greater microbe–plant symbiosis and diverse ecological niches. However, this study also observed that the soil microbiota under cover crops may be more primed for $N_2O$ production than bare soil under high nutrient availability. Thus, although cover cropping may effectively reduce the $NO_3^-$ leaching risk, further investigation is needed to understand the microbial contribution to $N_2O$ emissions under cover crop management. Overall, cover cropping has the potential to improve the soil health of a simplified cropping system by increasing its soil biodiversity, but its short-term use may have a limited impact in a heavily fertilized system. Future research should expand on this study by identifying bioindicators of similar taxonomic resolutions in various conditions and cropping systems, especially with longer use of cover cropping.

**Supplementary Materials:** The following supporting information can be downloaded at: https://www.mdpi.com/article/10.3390/agronomy12040954/s1. The Supplementary Material section has five tables and three figures. Table S1 shows the $\alpha$-diversity indices, observed number of amplicon sequence variants, Pielou's J, and Shannon's H', by N rate, CC, and their interactions for each bacteria, archaea, and fungi. Table S2 refers to the $\beta$-diversity measure and probability values (*p*- and q-value) of compositions of the bacterial, archaeal, and fungal communities by N rate, CC, and their interactions. Tables S3–S5 show the eigenvalue and cumulative proportion of the variability that each principal component (PC) explains in a dataset for each bacteria, archaea, and fungi, respectively; they also show the loading values that each indicator genus contributed to each PC. Figure S1 shows the location of the experimental site within the USA and the state of Illinois. Figure S2 shows the rarefaction curves of bacteria, archaea, and fungi. Figure S3 shows the estimated means of gene counts of bacterial indicators *Luteimonas* and *Pirellula* and fungal indicators *Albatrellus*, *Tetracladium*, and *Ajellomyces* normalized by central log-ratio transformation separated by the interactions of N rate and CC treatments.

**Author Contributions:** Conceptualization, M.B.V. and N.K.; methodology, N.K., M.A., C.W.R., M.C.Z., M.B.V., and S.L.R.-Z.; formal analysis, M.B.V. and N.K.; resources, M.B.V., S.L.R.-Z., and C.W.R.; data curation, M.B.V., M.C.Z., C.W.R., and N.K.; visualization, N.K.; writing—original draft preparation, N.K.; writing—review and editing, M.B.V., M.A., and M.C.Z.; supervision, project administration, and funding acquisition, M.B.V. All authors have read and agreed to the published version of the manuscript.

**Funding:** This research was funded by awards ILLU-802-978 and AG 2018-67019-27807, both from the United States Department of Agriculture, USDA-NIFA.

**Institutional Review Board Statement:** Not applicable.

**Informed Consent Statement:** Not applicable.

**Data Availability Statement:** The data have been deposited with links to BioProject accession number PRJNA809390 in the NCBI BioProject database.

**Acknowledgments:** We acknowledge Alvaro Hernandez and Mark Band from the Roy Carver Biotechnology Center at the Functional Genomics lab at the University of Illinois at Urbana-Champaign for their assistance in creating the amplicon libraries. We are thankful to Greg Steckel and Marty Johnson for their contribution in managing the experimental plots, and to Gevan Behnke for his assistance with soil sampling and overall lab management.

**Conflicts of Interest:** The authors declare that the research was conducted in the absence of any commercial or financial relationships that could be construed as a potential conflict of interest.

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
