# Peer review of "High-Resolution Indicators of Soil Microbial Responses to N Fertilization and Cover Cropping in Corn Monocultures"

_agronomy, doi:10.3390/agronomy12040954_

Round 1

Reviewer 1 Report

The introduction, results, discussion and conclusion all need major revisions. There are too many elements involved in the full manuscript, and there is a lack of a core point to connect them. Although the amount of relevant data seems to be large, there are too many repetitive indicators. I personally think that this is not an excellent manuscript and is not recommended for publication in this journal.

  1. It is suggested to revise a title that highlights the main theme of the full manuscript;
  2. The abstract is too long, the purpose of the experiment is not clear, there are too many unnecessary introductions to the experimental design, there is a lack of conclusive sentences, and the correlation with the topic is not strong
  3. The introduction has a total of 6 paragraphs, including the research background, cover cropping, soil microbiome, sequencing technology and bioinformatics, high taxonomic resolution indicators of CC and the goal of this study. There are too many points involved and cannot be found. The core part of the study in the full manuscript, and "the goal of this study" is best to be stated in separate articles. There should not be too many statements about the existing results, and the content that is not related to the content of the text needs to be merged with delete;
  4. The application of nitrogen fertilizer is involved in the manuscript, and it is best to supplement the indicators of soil physical and chemical properties and related nutrient elements content related to the test site in section 2.1;
  5. There is too much content in the headers of all the tables in the manuscript. The partially explained content can be placed at the bottom of the table as a note, and the letters (a, b, c) that appear in the table with significant differences are explained;
  6. The results only needs to describe the main results obtained in the manuscript, instead of repeating the same type of results;
  7. There are too many parts in the discussion about the statement of the results. During the discussion, explain and discuss the reasons for the main experimental results obtained (consistent with the previous results, why? Why are they inconsistent with the previous research results?), rather than the experimental results. After the discussion of the full manuscript is divided into 4 main parts, the content of 4.2 is divided again. It is not so much a discussion as a repetition of the result part, and the discussion needs to be rewritten;
  8. The conclusion is a summary of the main conclusions in the manuscript, not a simple superposition of all results, and needs to be rewritten

Reviewer 2 Report

Too long abstract. Contains too much methodological, experimental detail.

Introduction is also too long. L40-78 needs to be shortened slightly so that the introduction focuses on CC, N fertilization and microorganisms.

L105: explain “ITS”

L155-164: maybe add a map?

L185-191: how much soil was taken [g]? In how many repetitions? From which area?

Table 1: abbreviations should be explained in the footer under the table. This is how its title disappears. It should be explained at what n differences were determined and with what statistical test. In the table, it is worth standardising the number of digits after the decimal point in the numerical notations.

 L322-347: The description seems long and very detailed, making it a little difficult for the reader to understand.

Table 2: abbreviations should be explained in the footer under the table. This is how its title disappears.

L392: I do not see *, ** on the figure

Figure 2: cannot be of the same quality and style as Figure 1? The font on the Y axis is a bit blurred. It is worth improving the figure to make it look similar to Figure 1, which is coloured and better readable.

L405: I do not see *, **, *** on the figure

L430, 439, 486: I do not see *, *** on the figure

Figure 3, 4, 5, 6: it is worth improving the font - a little darker or bigger will be clearer and more readable. Especially when it comes to bacterial names.

454, 507: I do not see **, *** in the figure

Table 3: abbreviations should be explained in the footer under the table. This is how its title disappears.

The summary is long. It is worth extracting an unambiguous, short conclusion at the end.

Please check the literature list. In some places there are incorrect entries e.g. ref 61, 103.

Figure S01: the unit of depth on the X-axis should be given.

Figure S02: it would be good if the font was black on the Y-axis. It will be more readable.

Reviewer 3 Report

  1. Please be sure that the abstract and the conclusions section not only summarize the key findings of the work but also explain the specific ways in which this work fundamentally advances the field relative to prior literature.
  2. Please present a strong case for how this work is a major advance. This needs to be done in the manuscript itself, not just in the response to review comments. This is a very important point in terms of which
  3. The aim of the research is not clearly defined.
  4. Why the analyzes concern only one period in a year?
  5. Why the results of two years of microbiome studies were averaged?
  6. Table no. 1 is not readable. Too much information causes its illegibility.
  7. Indicate the possible risks of such research.
  8. Add your recommendations for future research.
  9. Make sure the references are added correctly according to the journal's instructions.
